# Ensuring Computers Understand Manual Operations in Production: Deep-Learning-Based Action Recognition in Industrial Workflows

**Zeyu Jiao [1]** **, Guozhu Jia [1] and Yingjie Cai [2],***

[1] School of Economics and Management, Beihang University, Beijing 100191, China;
raintbn@buaa.edu.cn (Z.J.); jiaguozhu@buaa.edu.cn (G.J.)

[2] Department of Electronic Engineering, The Chinese University of Hong Kong, Hong Kong 999077, China

\* Correspondence: caiyingjie96@163.com

**Abstract:** In this study, we consider fully automated action recognition based on deep learning in the industrial environment. In contrast to most existing methods, which rely on professional knowledge to construct complex hand-crafted features, or only use basic deep-learning methods, such as convolutional neural networks (CNNs), to extract information from images in the production process, we exploit a novel and effective method, which integrates multiple deep-learning networks including CNNs, spatial transformer networks (STNs), and graph convolutional networks (GCNs) to process video data in industrial workflows. The proposed method extracts both spatial and temporal information from video data. The spatial information is extracted by estimating the human pose of each frame, and the skeleton image of the human body in each frame is obtained. Furthermore, multi-frame skeleton images are processed by GCN to obtain temporal information, meaning the action recognition results are predicted automatically. By training on a large human action dataset, Kinetics, we apply the proposed method to the real-world industrial environment and achieve superior performance compared with the existing methods.

**Keywords:** deep learning; action recognition; convolutional neural network; spatial transformer network; graph convolutional network; industrial workflows

## 1. Introduction

Industrial production has seen dramatic technology transformations during recent decades with the development of emerging concepts and technologies, such as the Internet of things (IoT), cyber-physical systems (CPS), and big-data analytics (BDA). The automation and intelligence degree of industrial production have developed dramatically. Nevertheless, in some production processes, manual operations are still an integral part. In the current research, IoT, CPS, BDA, and other new technologies are difficult to extract manual operation data from, and there is no effective means for the analysis and processing of manual operations. However, the accurate recognition and processing of manual operations is of great significance to the human–computer interaction design [1], the improvement of product quality [2], and the reduction of costs [3] and is a necessary guarantee of the safety and security for the operators [4].

The main target of ensuring computers understand manual operations in production is to build a system to recognize the action category of humans in a period automatically based on the historical data of manual operations that allows computers. The source of the data is generally the manual operation videos captured by cameras in factories, which contain the poses of workers and the object being operated on over a continuous period of time, including both spatial and temporal information.

To find a way to extract the spatial and temporal information, action recognition has been greatly studied. Accurate action recognition is a highly difficult task [5] in industrial workflows principally for the following challenges: (1) Intra-class and inter-class differences. For the same action, the performance of different people may vary greatly. (2) Complexity of the environment, including occlusion, multiple shooting angles, illumination, low resolution and dynamic backgrounds. (3) Complexity in the temporal dimension. Human actions are often composed of multiple poses, and different starting points will affect the accuracy of action recognition enormously.

Motivated by the aforementioned challenges, we propose a multi-deep-learning model integrated spatial transformer graph convolutional network for action recognition that combines convolutional neural networks (CNNs), spatial transformer networks (STNs) and graph convolutional networks (GCNs) to recognize manual operations in industrial workflows. The spatial information in video is extracted by the CNN, the temporal information between the frames, and the spatio-temporal information is established through a GCN. To solve the problem that the human body presents different shapes under different camera angles, we use a STN to align the human body to enhance the recognition effect. At the same time, considering the difference in the workload of the upper and lower body of a human in industrial operations, we have added an attention mechanism to the model to adjust the weight of keypoints in action recognition, and improve the accurate identification of manual operations in industrial environments.

To validate the effectiveness of our method, we perform thorough experiments and achieve superior performance under both accuracy and false recognition rate metrics. The main contributions of the proposed method are as follows:

- We propose a novel method that combines multiple neural networks to recognize manual operations in an industrial environment, which is a rare attempt in the field.
- The proposed method leverages a spatial transformation network to reduce the recognition errors caused by the diversity of human poses in the real working environment.
- A graph convolutional neural network is constructed to extract the spatial and temporal information of the skeleton image at the same time, which, combined with the classifier, can accurately recognize human action.
- An attention mechanism. Considering the unique characteristics of the real-world production environment, different weights are applied to more than a dozen keypoints of the human body to improve the accuracy of recognition.

The remainder of this article is organized as follows. Section 2 introduces theories related to this study, including the pose estimation, graph neural network (GNN) and human action recognition. Section 3 details our proposed method. We present an empirical study to evaluate the model and discuss the relevant influencing factors in Section 4. Finally, in Section 5, we draw some conclusions and describe some future work that could be informed by this study.

## 2. Literature Review

### 2.1. Human Action Recognition

The main target of human action recognition is to recognize the category of human behavior over a period of time. According to different data sources, this category can be divided into image-based methods and methods based on other sensor data. Most of the studies about human action recognition focus on image sources, which generally use ordinary RGB (red-green-blue) cameras to record videos of human actions.

Prior to 2013, the methods [6–13] of action recognition mainly relied on algorithms to extract specific hand-crafted features in each frame, and then exploited pre-trained classifiers (e.g., support vector machine, SVM) to get the category of the frame, and finally, recognized actions by comparing the changes of actions between adjacent images. The quality of the hand-crafted features greatly

affects the performance of this kind of method. Some scholars have tried other modeling methods. Lv and Nevatia [14] leveraged a hidden Markov model (HMM) method to model temporal and causal relations between the different poses and activities, and used multi-class AdaBoost to classify the category.

Whitehouse et al. [15] also built an HMM-based model, in which the number of hidden states was set to the number of action classes. The improved dense trajectories (iDT) used by [7,8,13] is the method with the best effect, greatest stability, and highest reliability before deep learning enters the field of action recognition. The basic idea of iDT is to use the optical flow field to obtain some trajectories in video sequences, and extract features, such as the histogram of gray (HOF), histogram of oriented gradient (HOG), motion boundary histograms (MBH), and trajectory features along the trajectory. However, since the cost of calculation of optical flow is particularly high, the analysis efficiency of iDT remains extremely low (0.06 s for a pair of frames), and it would still introduce a bottleneck if done on-the-fly [16].

The latter methods [17–21] are closely related to the development of deep learning. The deep-learning-based method is far superior to the hand-crafted feature-based method in the recognition accuracy and processing speed of image and video. The mainstream deep-learning methods can be divided into CNN-based, RNN-based, and GCN-based methods according to the category of the model.

CNN-based methods have significant advantages in extracting spatial information from videos, and are easier for training, which has led many scholars to conduct research. Tu et al. [22] proposed a human-related multi-stream CNN (HR-MSCNN) architecture, which encodes appearance, action, and the captured tubes of the human-related regions to recognize human action. Ullah et al. [23] introduced a CNN-based method combined with an optimized deep autoencoder (DAE) to learn temporal changes of the actions, and classify actions based on SVM. Huang et al. [24] proposed a novel fusion network that combines temporal poses, spatial features, and action feature maps for classification by bridging the gap between the size differences between 3D and 2D CNN feature maps.

RNN-based methods have unique advantages at finding the time relationship between different frames in the video, and can find the temporal and spatial connection of human actions. Qi et al. [25] combined a spatio-temporal attention mechanism and semantic graph modeling to propose a novel attention semantic recurrent neural network for understanding human activities and individual behaviors in videos. Kuehne et al. [26] proposed a hybrid RNN-HMM approach with a hierarchical structure to solve the problem of weakly supervised learning of human actions from ordered action labels by constructing recognition in a rough to fine manner. Majd and Safabakhsh [27] proposed an extended version of the Long Short-Term Memory (LSTM) unit, which can perceive motion data as well as spatial features and time dependencies in the LSTM.

Although both CNN-based and RNN-based methods show certain advantages, when recognizing human actions, especially when combining key points on the human body to recognize the category of action, the relationship between human joints is more similar to a 2D vector than common image pixels. This also makes it possible to extract human motion information using GCN. Yan et al. [28] proposed a new spatio-temporal graph convolutional network (ST-GCN), a real-time spatial GCN model for solving human action recognition problems based on key points of the human skeleton. The excellent performance of GCN has inspired more scholars to explore in this direction. Shi et al. [29] further proposed a novel multi-stream attention-enhancing adaptive GCN that can learn uniformly or individually in an end-to-end manner based on input data. This data-driven approach increases the flexibility of models used for graph construction and brings more versatility to adapt to various data samples.

Other sensor data, such as wearable device data [30], RGB-Depth (RGB-D) video data [31], or textual instructions combined with sensor data [32], can be processed through effective methods and used to recognize human actions. However, since our method mainly focuses on image data, these methods are not reviewed in detail here.

## 2.2. Graph Convolutional Neural Network

A graph is a data structure that consists of a series of nodes and links between nodes (called edges). Unlike traditional images, which are composed of a series of neatly arranged pixels, the links between nodes in graph data can represent more relationships. For example, the links between human joint nodes can represent the human trunk and limbs. This laid the foundation for human action recognition using GCN, which borrows from the concept of convolution in CNN and can adaptively extract the information in the graph structure. Since Yan et al. [28] leveraged GCN to extract the spatio-temporal information of human action, compared with CNN-based and RNN-based methods, the effect has been significantly improved. Many scholars have paid attention to the excellent performance of GCN and the powerful expressiveness of graph structure.

Liu et al. [33] introduced a novel structure-induced graph convolutional network framework that defines a set of internal graphics for each input human skeleton. Then, to improve the performance of the action recognition, they developed an inter-graph model to model the relationship between different parts of the graph. Tang et al. [34] proposed a method based on deep progressive reinforcement learning to extract key frames, and used the GCN to capture the dependency between the joints for action recognition. The GCN demonstrates the ability to efficiently combine highly complex and non-differentiable rules, which is necessary to extract spatial and temporal information in the video.

The existing deep-learning-based action recognition methods are essentially based on CNN, RNN, or GCN or realize the recognition through a reasonable combination of the above three methods. Although some achievements have already been made, they pay little attention to operational recognition in industrial workflows, which is very important and urgent to study.

## 3. The Proposed Method

### 3.1. Overall Scheme

The overall scheme of our proposed framework is described in Figure 1. First, we extract frames from the input video and use the pose estimation method (described in detail in Section 3.2) to obtain the skeleton images of the human body. Each skeleton image corresponds to one frame of the video, and the coordinates in the width direction and the height direction of the image represent the position of the human body part in the space at the current moment. Also, the confidence of each keypoint in the pose estimation process is also entered the network as one of the features. Simultaneously, we consider the position of a frame on the timeline of the video, each point on the image has a coordinate in the temporal dimension.

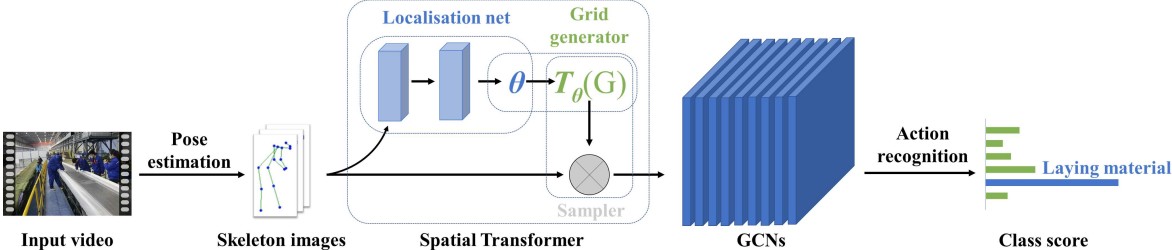

**Figure 1.** The overall framework for action recognition in industrial workflows. Graph convolutional network (GCN).

The coordinates of the joint in the skeleton image (including 2 spatial coordinates, 1 confidence parameter, and 1 temporal coordinate) and the connection relationship between the joints are used as input of the model we proposed. Then, the STN adaptively adjusts the human body pose by translating, rotating, and scaling the skeleton image to eliminate the influence of the human body pose diversity. Finally, the high-dimensional features in skeleton images are automatically extracted by the multi-layer GCNs, and the class score of action categories can be obtained, thereby realizing

the recognition of the manual operation in the industrial workflows. The recognition process can be expressed as Algorithm 1.

---

**Algorithm 1** The process of action recognition

---

**Input:** $V$, Video to be recognized, pre-trained parameters $\theta_{ij}(i = 1, 2, j = 1, 2, 3)$ of the STN
**Output:** Result matrix $O_{n-i}[o_0, o_1, \cdots, o_{n-i}]$, $o_i(i \in [0, n-1])$ is the probability of operation $i$
1: Extract each frame of video $V$ to get a series of images $I_j$, where $j$ represents the order of the frames
2: Detect people in each image
3: Converting images $I_j$ into skeleton images $S_j$ through pose estimation, with the order $j$ unchanged
4: Affine and scale transformations for $S_j$ using trained parameters $\theta_{ij}$
5: Input the transformed skeleton graph $S_j$ into GCN in order $j$ to obtain human action information
6: Feed the information extracted by GCN to the SoftMax classifier and get the result matrix $O_{n-i}$

---

The components in the method we proposed will be elaborated in detail below.

### 3.2. Pose Estimation

Referring to action recognition, let us first review how humans recognize an action. Usually, the recognition of another person's actions is achieved by observing another person's continuous pose over a period of time. Like figure skating, the referee evaluates the athlete's movement by scoring some specific pose of the athlete over a period. Therefore, following the process of human recognition, to recognize the manual operation in the industrial environment, it is first necessary to know where the person is and his (or her) pose at a certain moment. The entire process of human pose estimation is shown in Figure 2.

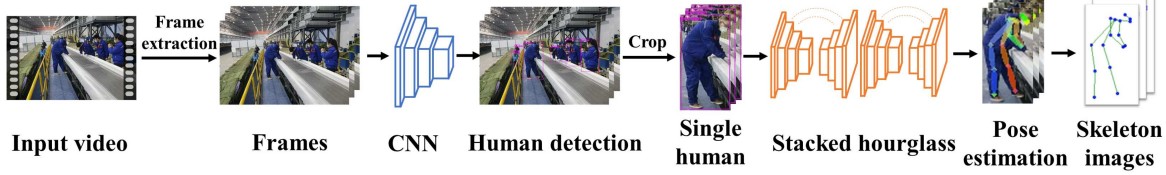

**Figure 2.** The entire process of human pose estimation. Convolutional neural network (CNN).

### 3.2.1. Human Detection for Pose Estimation

To balance the accuracy and speed of detection, we used the You Only Look Once version 3 (YOLOv3) [35] model for human detection. By extracting each frame of the input video and resizing it to the appropriate size, the human in the video is being detected by the YOLOv3 model and marked with a rectangular box, called the bounding box. For a single person in the video, the model's output contains five predicted values: $x, y, w, h$, and $confidence$. $x$ and $y$ are the center coordinates of the bounding box, $w$ and $h$ represent the width and height of the bounding box, and confidence represents the confidence that an object in the bounding box is human. The detection of the human body can be achieved by filtering out the bounding boxes whose confidence is below a certain threshold.

### 3.2.2. Pose Estimation for a Single Human

Based on human detection, a single human body in the video can be cropped out according to the coordinates of the bounding box $(x, y, w, h)$. Here we leverage the stacked hourglass model (SHM) [36] for single human pose estimation. The SHM uses a full CNN to output precise pixel locations of keypoints of the human body in cropped images, and multi-scale features are employed to capture spatial position information of various joint points of the human body. The single hourglass network structure is shaped like an hourglass, and the top-down to bottom-up is repeated to infer the position of the joint point of the human body. Each top-down to bottom-up structure is a stacked hourglass

module. As the SHM only involves convolution calculations, it essentially only needs to complete the multiplication of the matrix, which also ensures the speed of the pose estimation.

### 3.3. Spatial Transformer Networks

After human detection and pose estimation, the input video data is converted into a sequence of skeleton images. In an industrial production environment, people may appear anywhere in the field of view, presenting different poses and sizes, which causes some interference in the recognition of manual operations. To make our method more robust to the environment and reduce the impact of the diversity of human poses on the recognition results, we employ an STN to adaptively adjust each frame, and the adjustments include translation, rotation, and zoom.

According to Jaderberg et al. [37], any affine transformation process can be described using six parameters. Therefore, the STN achieves adaptive correction of human poses in industrial environments by regressing these six parameters. The structure of the STN is shown in the spatial transformer part of Figure 1. Differently from manually correcting the pose of the human body in each frame or directly using the CNN to extract features without any pre-processing, the STN consists of three parts: the localization net, grid generator, and sampler.

The localization net is essentially a regression network, the input of which is the skeleton image, and the output is the six parameters of the affine transformation. The structure of the localization net is usually a fully connected network or a convolutional network followed by a regression layer to train the parameters, which is denoted as $\theta$ in Figure 1. The grid generator (i.e., $T_\theta(G)$ in Figure 1) transforms the skeleton image by taking advantage of the parameter $\theta$, and calculates the coordinates in the original skeleton image corresponding to each position in the target image. The generation process can be expressed as follows:

$$\begin{pmatrix} x_i^s \\ y_i^s \end{pmatrix} = \mathcal{T}_\theta\left(G_i\right) = \theta \begin{pmatrix} x_i^t \\ y_i^t \\ 1 \end{pmatrix} = \begin{bmatrix} \theta_{11} & \theta_{12} & \theta_{13} \\ \theta_{21} & \theta_{22} & \theta_{23} \end{bmatrix} \begin{pmatrix} x_i^t \\ y_i^t \\ 1 \end{pmatrix} \tag{1}$$

where $\left(x_i^t, y_i^t\right)$ is the coordinates of a position on the target image and $\left(x_i^s, y_i^s\right)$ is the coordinates of the corresponding position on the input skeleton image. The *sampler* samples the skeleton image according to the coordinate information in $T_\theta(G)$, and copies the pixels to the target image. However, $\left(x_i^s, y_i^s\right)$ always falls in the middle of several pixel points of skeleton image, so in this study we use bilinear interpolation to calculate the gray value corresponding to this point. Most importantly, according to the research in Jaderberg et al. [37], the gradient can be transmitted during the STN transformation process, so end-to-end training can be continuously performed in the network to correct the parameters.

In summary, the training result of STN is six parameters, and the skeleton images can adaptively perform affine transformation after STN. In the case where the input data has a large spatial difference, this network can be added to the existing convolutional network to improve the accuracy of classification.

### 3.4. Graph Convolutional Neural Network

After the correction of STN, the skeleton images are adjusted to the appropriate size and pose, and the input video is also converted into a series of images at different positions on the time axis. As shown in Figure 3, each blue dot represents a keypoint on the human body, the green lines represent the human limbs and the torso, and the orange line represents the connection of the same keypoint in consecutive frames. Human skeleton images at different moments form an interconnected network in 3D space, called a spatial temporal skeletal graph, the three dimensions of which are the width direction and height direction of the image and the temporal dimension, so any keypoint can be represented as a space coordinate $(w, d, c, t)$. $w$ and $d$ correspond to the position of the keypoint in the skeleton image, $c$ represents the confidence of this keypoint, $t$ corresponds to the moment

corresponding to the keypoint. Therefore, the remaining question is how to extract the spatial and temporal information from these keypoints and the connections between them to recognize actions.

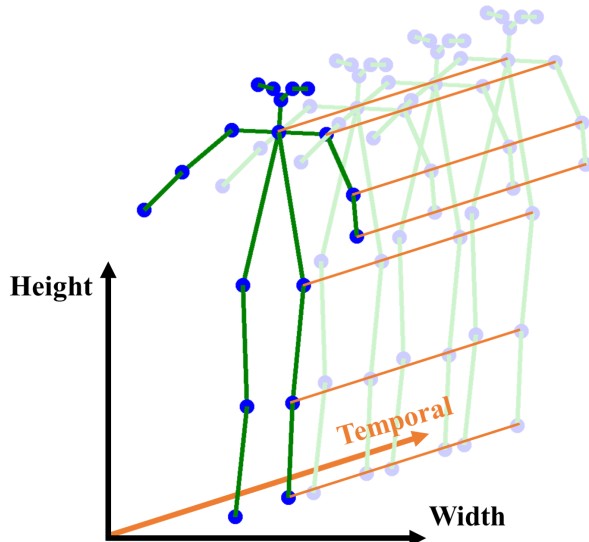

**Figure 3.** Model structure of stacked hourglass model (SHM).

Here we use the spatial temporal graph convolution network [28] to form a hierarchical representation of the human skeleton. For a video with a total of $T$ frames, the spatial temporal skeletal graph with $N$ keypoints can be represented as $G = (V, E)$, where $V = \{v_{ti}|t = 1, 2, \ldots, T, i = 1, 2, \ldots, N\}$ denotes the keypoint set of the graph, $E$ represents the line between the keypoints in the spatial temporal skeletal graph, composed of two subsets of $E_S$ and $E_F$. $E_S = \{v_{ti}v_{tj}|(i, j) \in H\}$ is the connection of the keypoints in each frame, $H$ represents a collection of keypoints of the human body, $E_F = \{v_{ti}v_{(t+1)i}\}$ represents a link between keypoints in consecutive frames, and each element in $E_F$ represents a trajectory of a particular joint over time. In terms of spatial information, according to Yan et al. [28] and Kim et al. [38], the graph convolution in single frame can be expressed as

$$f_{\text{out}}(v_{ti}) = \sum_{v_{tj} \in B(v_{ti})} \frac{1}{Z_{ti}(v_{tj})} f_{in}(v_{tj}) \cdot \mathbf{w}(v_{ti}, v_{tj}) \tag{2}$$

where $f_{in}$ and $f_{out}$ are the input and output of the graph convolution, $B_{v_{ti}}$ is the collection of 1-distance neighbors of the target keypoint $v_{ti}$, $Z_{ti}(v_{tj})$ is used to balance the contribution of different subsets to the output, which equals the cardinality of the corresponding subset, and $w$ is the weight function, which is used to compute the inner product of the input skeleton image.

For the fusion of spatial and temporal information, suppose that $A$ is an adjacency matrix of $N \times N$, which represents the connection relationship between keypoints between adjacent frames. However, unlike ordinary images, if an adjacent matrix is used to define a neighborhood, the number of keypoints in the neighborhood of each keypoint is not fixed. This makes it difficult for us to determine: (1) The parameter dimensions of the convolution kernel that need to be used. (2) If the weight matrix is aligned with the keypoints in the neighborhood for inner product operations.

Here we have designed some strategies similar to Yan et al. [28]: (1) Suppose $v_{tg}$ is the barycenter of the human body in skeleton images. (2) For any keypoint $v_{ti}$, the keypoints connected to it form a neighbor set, the keypoint $v_{ti}$ is called the root node of this neighbor set. (3) In the neighbor set of any root node, all points with a distance greater than the barycenter from the root node are called

centrifugal subsets, and all points with a distance less than the barycenter from the root node are called centripetal subsets. According to the research in Kim et al. [38], Equation (2) can be converted into

$$f_{\text{out}} = \sum_j \Lambda_j^{-\frac{1}{2}} \mathbf{A}_j \Lambda_j^{-\frac{1}{2}} f_{in} \mathbf{w}_j \tag{3}$$

where $f_{in}$ is the input feature map, which can be represented by a tensor $(N, T, C)$ dimensions, and $C$ is the number of input channels. $A_j$ is a set of adjacency matrix subsets. When $j = 0$, $A_j$ represents each keypoint self-joining. When $j = 1$, $A_j$ represents the connection of centripetal subset. When $j = 2$, $A_j$ represents the connection of centrifugation subset. $W_j$ is a weight matrix formed by superimposing the weight vectors of multiple output channels. $\Lambda_j$ can be calculated as $\Lambda_j^{ii} = \sum_k \left( A_j^{ki} \right) + b$, $b$ is an offset parameter, initially set to 0.001 to avoid $\Lambda_j$ being an empty set. In this study, we used a network structure with a nine-layer GCN to extract spatial and temporal information in the skeleton image, where the first three layers have 64 channels for output, the following three layers have 128 channels for output, and the last three layers have 256 channels for output. The parameters in this model can be updated through end-to-end training.

*3.5. Attention Mechanism*

In the actual working environment, the workload of the upper body is often more than that of the lower body. If the keypoints in the spatial temporal skeletal graph have the same weight under different workloads, it may make the recognition results worse. Therefore, the attention mechanism is introduced in our model, and the weights of different keypoints are adaptively allocated by stacking the basic graph self-attention layer. The basic graph self-attention layer can assign different weights to the neighbor set of the root node, indicating the different importance of the neighbor sets of the root node. First, the attention coefficient between any two keypoints can be defined as

$$E_{ij} = a \left( v_{ti}, v_{tj} \right) \tag{4}$$

where $a$ indicates a shared attention mechanism. In general, the relationship between any two keypoints can be calculated by the attention mechanism, so all the keypoints can be represented by the attention coefficient of one keypoint. However, based on the spatial similarity assumption, a keypoint is closely related to the keypoint within a certain range, and when there are many keypoints, the calculation amount is very large. Therefore, for $v_{ti}$, only the keypoints in the neighbor set are used to calculate the attention coefficient.

When only the keypoints in the neighbor set of $v_t i$ are considered, Equation (4) can be rewritten as

$$\alpha_{ij} = \text{softmax}_j \left( E_{ij} \right) = \frac{e^{E_{ij}}}{\sum_{k \in N} e^{E_{ik}}}. \tag{5}$$

Here, for the convenience of calculation, the SoftMax function is used to normalize the attention coefficient. The single-layer feedforward neural network, which is parameterized by the weight vector, represents the attention mechanism $a$, and the *LeakyRuLU* is adopted to activate the node, so the calculation formula is specifically expressed as

$$\alpha_{ij} = \frac{e^{LeakyRuLU\left[a^T \left( v_{ti}, v_{tj} \right)\right]}}{\sum_{k \in N} e^{LeakyRuLU\left[a^T \left( v_{ti}, v_{tk} \right)\right]}}. \tag{6}$$

According to the study of Veličković et al. [39], each node calculates its new representation based on the attention weight coefficients of its neighbor set and the keypoint in neighbor set, which can be expressed as

$$\alpha_{ii} = \delta \left( \sum_{j \in N} \alpha_{ij} v_{tj} \right) \tag{7}$$

where $\delta$ represents a nonlinear transformation. Finally, after the K-th update of the attention coefficient, the average value under each attention mechanism is calculated as

$$\alpha_{ii} = \delta \left( \frac{1}{K} \sum_{k=1}^{K} \sum_{j \in N} \alpha_{ij}^K v_{tj} \right). \tag{8}$$

## 4. Experimental Study

In this section, we evaluate the performance of the proposed method in real industrial workflows. We experimented with the proposed method in recognizing six basic manual operations, including blasting sand, spraying gelcoat, laying materials, pumping gas, plastering adhesive, and using a remote controller, all in the wind turbine blade manufacturing process. All video data is obtained using a normal RGB camera characterized by 1/2.7" complementary metal oxide semiconductor (CMOS) sensor with 12 megapixels. The details of the dataset are shown in Section 4.1. The implementation details of the method we proposed are given in Section 4.2. The experimental results and comparisons with existing studies are given in detail in Section 4.3. In Section 4.4, we discuss the role of each module in the model and its impact on the experimental results.

### 4.1. Datasets

When constructing the dataset in the actual industrial scene, we follow the form of the Kinetic dataset [40], dividing the six basic operations video data into 10 s of video and converting the frame rate to 30 frames per second (FPS), so that each video in the manual operation dataset consists of 300 images. As mentioned in the *Introduction*, there are three main challenges in action recognition: (1) intra-class and inter-class differences; (2) complexity of the environment; and (3) complexity in the time dimension. Challenge 2 has been overcome by using the skeleton image. When building our dataset, we collected multiple videos of the same person performing the same operation to reduce the impact of intra-class differences on recognition accuracy. At the same time, we collected videos from multiple groups of people performing the same operation to reduce the impact of inter-class differences on recognition accuracy. In addition, when dividing the video, we selected multiple starting points of the same action, which can increase the number of training samples on the one hand, and reduce the influence of different starting points on the accuracy of action recognition to some extent.

During training, the video dataset is randomly divided into three subsets of train, validation (val), and test. We use a pattern similar to Kinetic to fix the number of val and test sets to 50 and 100. The amount of video data for each operation and the partitioning results are shown in Table 1.

**Table 1.** The amount of video data for each operation and the partitioning results.

| Operation | Train | Val | Test | Total |
|---|---|---|---|---|
| Blasting sand | 552 | 50 | 100 | 702 |
| Spray gelcoat | 599 | 50 | 100 | 749 |
| Laying materials | 662 | 50 | 100 | 812 |
| Pumping gas | 605 | 50 | 100 | 755 |
| Plastering adhesive | 803 | 50 | 100 | 953 |
| Using remote controller | 728 | 50 | 100 | 878 |

### 4.2. Implementation Details

Although our dataset is representative, the total number of videos is quite limited compared to the kinetic dataset. When constructing the entire model, we used a two-stage training method to independently train the pose estimation model and the skeleton-based action recognition model.

In human detection, the YOLOv3-416 model (the input size is 416 × 416) and YOLOv3 default weight are adopted, but are different from the confidence threshold applied in the original model, which defaults to 0.25. Through experiments, we select the best confidence threshold in the industrial environment, and only output the results when the confidence is greater than 0.4. Then we leverage the stacked hourglass network to obtain the 18 keypoints in each frame and form a sequence of skeleton images.

When using our method to recognize actions, we first pre-train our network based on the Kinetic dataset, and then use the datasets of the six operations to adjust the network weights. The residual mechanism proposed by He et al. [41] is applied so that the neural network can learn the residual of the previous network output without learning the entire output, which can reduce the gradient disappearance and information loss during deep network training. For STN, ResNet-18 [41] is applied as a localization net. We set the dropout probability in each layer of GCN to 0.5 to avoid over-fitting the model.

Finally, we use the SoftMax classifier to convert the output of the network into the probability value of the corresponding operation, and select the one with the highest probability as the voting result. Throughout the training process, the learning rate is initialized to 0.01 and decay to 0.1 after every 10 epochs. The adaptive moment estimation (Adam) optimization algorithm is used for end-to-end training of the entire model. All experiments were conducted on a PyTorch deep-learning framework with an Intel i7-6700K CPU at 4.0 GHz with 8 GB RAM and a GTX1080Ti GPU with 16 GB memory.

### 4.3. Experimental Results

To evaluate the performance of the human detection and pose estimation models, we extracted 1200 frames from the videos in train set, including 200 per operation. These images contain a total of 1316 human images, and we manually annotate 18 keypoints of the humans according to the format of the Microsoft COCO dataset.

The evaluation results of human detection are shown in Table 2. Here we consider the effect of different thresholds on the test results.

**Table 2.** The results of the human detection (considering different thresholds).

| Threshold | Precision (%) | Recall (%) | F-Measure (%) | G-Mean (%) |
|-----------|---------------|------------|---------------|------------|
| 0.15 | 73.80 | 99.54 | 84.76 | 85.71 |
| 0.2 | 76.68 | 99.47 | 86.60 | 87.34 |
| 0.25 | 78.16 | 99.01 | 87.36 | 87.97 |
| 0.3 | 88.62 | 98.78 | 93.42 | 93.56 |
| 0.35 | 93.64 | 98.40 | 95.96 | 95.99 |
| 0.4 | 95.34 | 97.87 | 96.59 | 96.60 |
| 0.45 | 95.92 | 91.19 | 93.49 | 93.52 |
| 0.5 | 96.56 | 76.75 | 85.52 | 86.09 |
| 0.55 | 97.86 | 72.95 | 83.59 | 84.49 |
| 0.6 | 98.08 | 69.91 | 81.63 | 82.81 |

The *Precision* in the test results is obtained by calculating the ratio of the correctly detected quantity to the total predicted quantity. The *Recall* is obtained by calculating the ratio of correctly detected to 1316 people in these images. F-measure and G-mean are computed based on Equations (9) and (10).

$$F - \text{measure} = \frac{2 \times \text{Precision} \times \text{Recall}}{\text{Precision} + \text{Recall}} \tag{9}$$

$$G - mean = \sqrt{\text{Precision} \times \text{Recall}}. \tag{10}$$

After the humans in images are detected, we evaluate the results of the pose estimation. When the predicted keypoints and the corresponding manually annotated keypoints are within a certain distance range, and the connection between the points is correct, the pose estimation is considered correct. Table 3 shows the results of the pose estimation of each operation.

**Table 3.** The results of pose estimation of each operation.

|  | Blasting Sand | Spraying Gelcoat | Laying Materials | Pumping Gas | Plastering Adhesive | Using Remote Controller | Average |
|---|---|---|---|---|---|---|---|
| Accuracy | 96.5% | 92.5% | 91.0% | 93.5% | 93.0% | 90.5% | 93.6% |

We also evaluate the performance of the entire model in the recognition of the actions and compare it with the existing research. As shown in Table 4, the video used in the test is derived from real industrial production, and the mainstream methods in the field of action recognition are used for comparison. Figure 4 shows the confusion matrix of the results of our method on the test data. At the same time, we visualize the results in Figure 5, showing the specific implementation process and corresponding results.

**Table 4.** Comparison of our methods and existing methods on test data.

| Methods | Blasting Sand (%) | Spraying Gelcoat (%) | Laying Materials (%) | Pumping Gas (%) | Plastering Adhesive (%) | Using Remote Controller (%) | Total (%) |
|---|---|---|---|---|---|---|---|
| iDT+SVM [42] | 75.26 | 83.51 | 72.16 | 80.41 | 82.47 | 79.38 | 76.50 |
| Two-stream [16] | 82.47 | 85.57 | 83.51 | 77.32 | 78.35 | 91.75 | 80.83 |
| C3D [43] | 85.57 | 90.72 | 90.72 | 90.72 | 87.63 | 61.86 | 82.17 |
| ST-GCN [28] | 85.57 | 92.78 | 92.78 | 91.75 | 88.66 | 87.63 | 88.33 |
| R(2+1)D [44] | 88.66 | 93.81 | 93.81 | 91.75 | 90.67 | 92.78 | 90.67 |
| Ours | 92.78 | 95.88 | 94.85 | 93.81 | 89.69 | 96.91 | 91.17 |
| W/o attention | 85.57 | 86.60 | 87.63 | 88.66 | 89.69 | 90.72 | 88.50 |
| W/o STN | 82.22 | 88.92 | 84.79 | 85.05 | 85.57 | 80.41 | 81.17 |

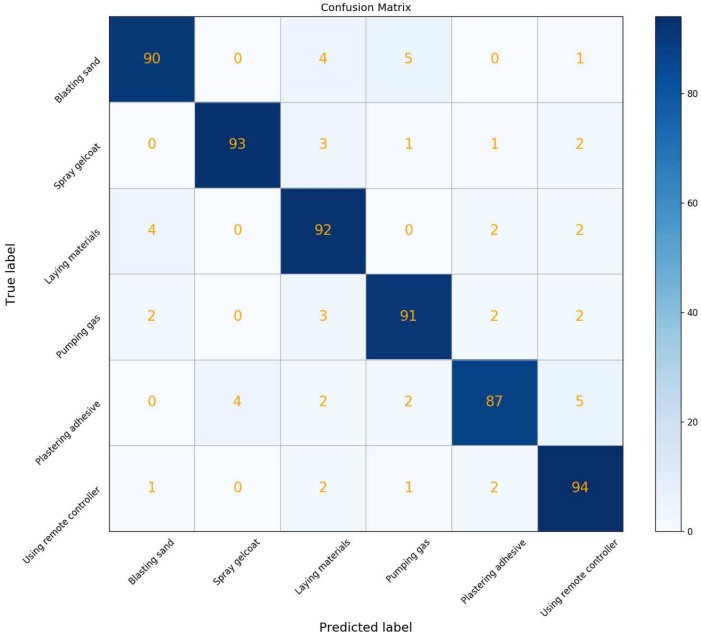

**Figure 4.** Confusion matrix of the results of our proposed action recognition method.

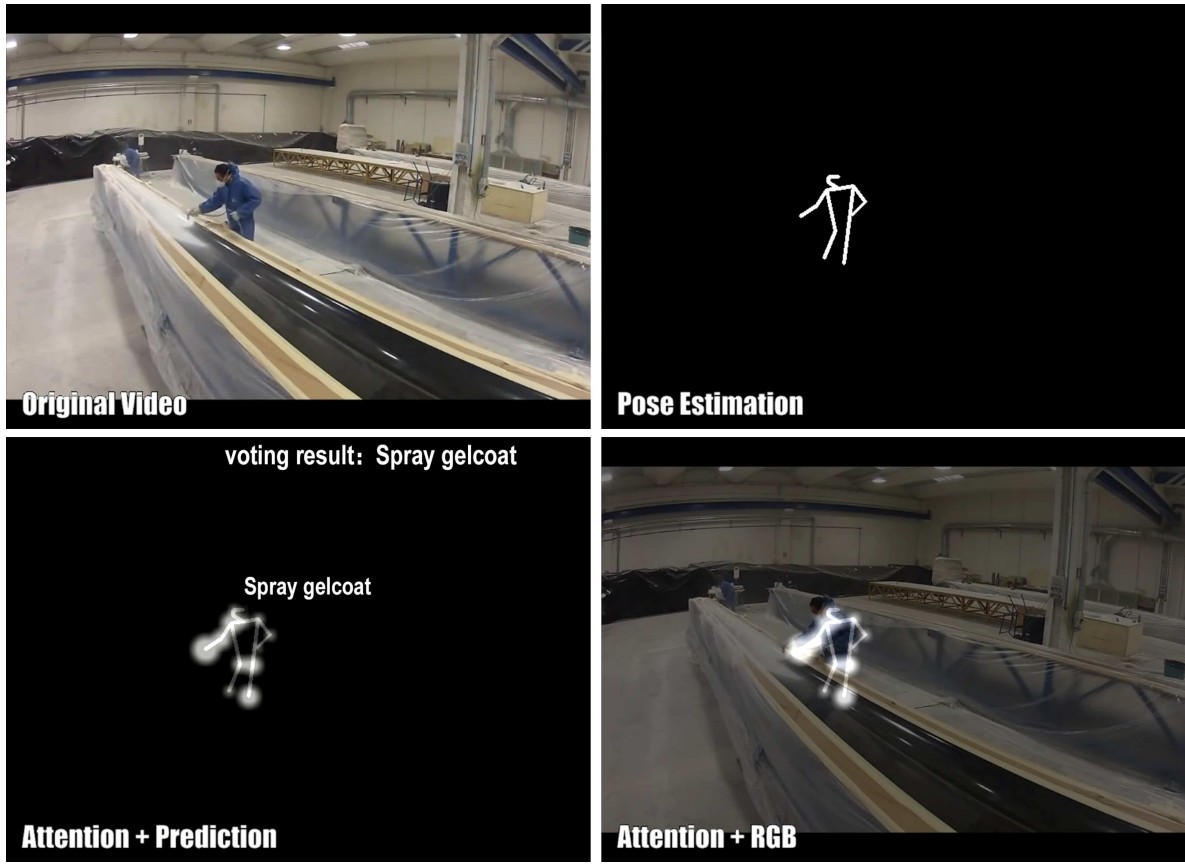

**Figure 5.** Visualization of the results of the method we proposed.

In the end, we conducted 100-hour field tests of the existing methods shown in Table 4 in 3 different workshops of China UnitedPower Company with three different cameras, respectively. Table 5 shows the final false recognition rate of the field tests, which show that our method significantly outperforms the existing methods.

**Table 5.** The results of field tests in real workshops.

|  | iDT+SVM [42] | Two-Stream [16] | C3D [43] | ST-GCN [28] | R(2+1)D [44] | Ours |
|---|---|---|---|---|---|---|
| False recognition rate (%) | 36.5% | 32.5% | 21.33% | 14.56% | 18.93% | 9.50% |

### 4.4. Discussion

Among the methods we propose, the STN and attention mechanism are two core innovations that play a key role in action recognition in industrial workflows. In this section, we review the mechanisms of these two parts and their impact on the results.

As described in Section 3.3, the STN is used to adaptively adjust the pose and size of the humans in the skeleton images. Taking the operator in Figure 5 as an example, when performing the spraying gelcoat operation, he needs to face the fan blade or walk back and forth along the blade, which causes the position of the keypoints on the human body to be in a rotating state in the space, which makes it difficult to extract spatial and temporal information using the GCN. By the same token, when the operator is at the top of the blade and the bottom of the blade, different sizes will appear in the video. Without the adjustment of the STN, the same action can lead to different spatial information, resulting in erroneous recognition.

We also experimentally tested the effect of removing the STN on the results, as shown by our method without STN in Table 4. Using STN significantly improved the performance of our method by

nearly 10%. Considering the different influences of different keypoints of the human images on the action recognition results in the actual production environment, the attention mechanism is necessary for accurate motion recognition. As also shown in Figure 5, we visualize the weights of different keypoints contributing to the final result, and use the shadow size near the keypoints to indicate the difference in weight. Our method considers that the action in the video belongs to the spraying gelcoat category, which is mainly determined by the keypoints on the right hand of the operator.

The attention mechanism automatically assigns different weights to different keypoints to make our model more in line with actual industrial production. We also tested the effect of removing the attention mechanism on the results, as shown by our method without attention mechanism in Table 4.

In addition, we also study the effect of pose estimation on the results. As shown in Table 6, we compared the accuracy of using different pose estimation methods and the impact on action recognition results. The experimental results show that different pose estimation methods have an impact on the accuracy of constructing the skeleton image; however, with the use of dropout and other methods in subsequent models, our model is robust to skeleton data and is not susceptible to subtle changes. Therefore, when the accuracy difference is small, we choose the fastest processing method to recognize the action in the video more quickly. In the industrial workflow, our method leaves a certain margin for the difference in performance of the same kind of operation, which also makes our method more suitable for actual production.

**Table 6.** Comparing the effects of different pose estimation methods.

| Methods | Pose Estimation Accuracy (%) | Action Recognition Accuracy (%) | Speed (FPS) |
|---|---|---|---|
| OpenPose [45] | 89.8 | 91.00 | 9.9 |
| AlphaPose [46] | 95.1 | 91.67 | 20.2 |
| YOLOv3+SHM | 93.6 | 91.17 | 23.3 |

## 5. Conclusions and Future Work

In this study, we propose a novel deep-learning-based action recognition method that can automate and accurately recognize manual operations in industrial workflows. Our main contributions are summarized as follows. First, we propose a new framework that uses human detection and pose estimation to obtain the skeleton image. Based on the skeleton image, automatic recognition of the manual operation can be realized. Second, we explore the use of STN to correct skeleton images to avoid the impact of complex real-world production environments on the recognition results. Third, we introduce a GCN to simultaneously extract spatial and temporal information in skeleton images. Fourth, an attention mechanism is added to our method to make the method more applicable to the actual production process and improve the accuracy of the recognition. The results of our method on real production video outperform the existing research, and our study is a kind of next generation industrial and manufacturing technology, which may be significant for the industrial revolution in the coming decades.

It should be noted that only the actions in 2D RGB video are recognized in this study, but the human operation is done in a complex 3D space. The action recognition method based on the 3D skeleton image is more in line with the actual production requirements, and can be recognized more accurately. How to acquire 3D skeleton images and how to recognize actions based on 3D skeleton images is the main work of the future.

**Author Contributions:** Z.J. designed the research method, contributed to the experimental section and wrote the draft. G.J. and Y.C. gave a detailed revision and provided important guidance. All authors have read and agreed to the published version of the manuscript.

**Funding:** This research was funded by the National Natural Science Foundation of China (Grant number 71772010) and Technical Foundation Research Project of Ministry of Industry, Information Technology of PRC (Grant number JSZL2016601A004).

**Conflicts of Interest:** The authors declare no conflict of interest.

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
