# Peer review of "Ensuring Computers Understand Manual Operations in Production: Deep-Learning-Based Action Recognition in Industrial Workflows"

_applsci, doi:10.3390/app10030966_

Round 1
Reviewer 1 Report
The paper presents a human activity recognition pipeline targeting manual operations in an industrial environment. The authors use a combination of well known deep learning techniques for human detection, pose estimation, pose correction and activity recognition. Extensive results are presented on a moderate dataset and comparison with other techniques is performed. Thus the contribution of the work lies on the combination of the specific techniques used and their application o a specific environment.
The paper should be significantly improved before is acceptable for publication. The introduction should clearly explain the motivation and scope of the work. Although an industrial environment presents some challenges it can also be controlled (e.g. use proper illumination, place cameras at required locations). Why are the proposed techniques needed?
The literature review is poorly written. More detailed explanation of past work is required highlighting their key ideas and their shortcoming also with respect to the specific application.
The rest of the paper describes the building blocks of the classifier. We suggest that the authors start with a high level description of the algorithm (thus suitable for a non expert) and then focus only on the inputs and outputs of the algorithm and any adapation performed in the paper (e.g. addition of layers). Figure 4 and figure 5 for example are of no value to the paper, and affine transformation descriptions (eq.1-4) are also superficial and well known. Also important information is needed (e.g. in 2.4 what is the output of the network).
In the experiments the authors should focus on the challenges of industrial environment as claimed. For example although inter-class and intra-class variability is mentioned I see no experiment proving that the proposed scheme can do better in this respect.
Author Response
We sincerely appreciate your hard work and thoughtful suggestions that have greatly improved this article. Below is a point-by-point response to your comments:
1. The introduction should clearly explain the motivation and scope of the work. Although an industrial environment presents some challenges it can also be controlled (e.g. use proper illumination, place cameras at required locations). Why are the proposed techniques needed?
Thank you for your valuable suggestions, it is indeed possible to control the industrial environment in a certain factory by the proper method as you said. The challenge we are talking about is not a challenge in a single factory, but we want to find a universal method that can be applied to different environments in different factories. This method needs to be more robust to illumination and the environment. It can be guaranteed to be reliable when applied in different environments in different factories. After all, it is difficult to ensure that the illumination and camera angles in each factory are the same in real production.
2. The literature review is poorly written. More detailed explanation of past work is required highlighting their key ideas and their shortcoming also with respect to the specific application.
Thank you for your suggestion, we have carefully rewritten the literature review and described the past work in detail.
3. The rest of the paper describes the building blocks of the classifier. We suggest that the authors start with a high level description of the algorithm (thus suitable for a non expert) and then focus only on the inputs and outputs of the algorithm and any adaptation performed in the paper (e.g. addition of layers).
Thank you for your valuable suggestions. We have added a high-level description of the algorithm in the method section. The algorithm can be found in Section 3.1.
4. Figure 4 and figure 5 for example are of no value to the paper, and affine transformation descriptions (eq.1-4) are also superficial and well known. Also, important information is needed (e.g. in 2.4 what is the output of the network).
Thank you for your valuable suggestions. Figure 4, figure 5 and Eq. 1-4 really do not have much value. We have removed them in the revised manuscript. At the same time, in order for readers to better understand the role of different sections, we have explained the output of networks and the meaning of existence at the end of each subsection.
5. In the experiments the authors should focus on the challenges of industrial environment as claimed. For example although inter-class and intra-class variability is mentioned I see no experiment proving that the proposed scheme can do better in this respect.
Thank you for your valuable suggestions. We explained in 4.1 that when building our dataset, we collected multiple videos of the same person performing the same operation to reduce the impact of intra-class differences on recognition accuracy. At the same time, we collected videos from multiple groups of people performing the same operation to reduce the impact of inter-class differences on recognition accuracy. In addition, when intercepting the video, we select multiple starting points of the same action, which can increase the number of training samples on the one hand, and reduce the influence of different starting points on the accuracy of action recognition to some extent.

Reviewer 2 Report
The paper proposes an neural network approach to recognising human activities in industrial settings. They combine CNN for human detection and pose estimation, STNs for adjusting the pose for more robustness, and GCN for modelling spatial and temporal relations in the change of pose. The approach is compared to state of the art approaches and shows that it performs comparable to them.
Generally the paper is interesting but I am still not sure what is the advantage compared to existing approaches as the empirical evaluation shows that it performs comparable to state of the art approaches. I believe that the authors should show more clearly what is the advantage of their approach. Apart from that, below are some more comments:
- In the literature review, when talking about graph neural networks, the authors are mixing networks applied to different input data but they do not make clear distinction between approaches applied to images and approaches applied to some other data (e.g. sensor data). As the whole literature review centers around image based AR, this sudden shift to other types of data is confusing.
- In the literature review, to make the review complete the authors should also address other types of data used for AR in industrial settings. For example there are works using sensors (e.g. see [1]) and as sensors are considered less intrusive than camera, the authors should at least mention this type of research.
- Also in the literature review, the authors talk about classical ML approaches for AR but they don't mention approaches such as HMMs, which model temporal and causal relations between the different poses and activities (e.g. see [2]). I think that such approaches are extremely relevant as there are certain causal and temporal relations between human activities especially in goal oriented scenarios (and tasks in industrial settings are usually goal oriented). I believe the authors should mention the existence of such approaches and how they differ from the way their approach utilises temporal and spatial information.
- Also along these lines, I wonder if the authors are aware of approaches that use high level textual instructions describing the way tasks should be executed in order to enhance the recognition performance. For example, [3] generates models of human behaviour from textual instructions such as manuals. These models are then combined with sensor data to perform activity recognition.
- In the evaluation part, the authors compare their approach with different state of the art approaches. To me the results seem very similar. The authors should perform statistical tests to test if their approach significantly outperforms the existing approaches.
- General comments:
= The abstract and introduction have some language problems and need to be proofread. The language in the rest of the paper is fine.
= The section numbering starts with 0 instead of with 1
[1] F Moya Rueda et al. Convolutional neural networks for human activity recognition using body-worn sensors. Informatics 5 (2), 26. 2018
[2]S Whitehouse et al. Evaluation of cupboard door sensors for improving activity recognition in the kitchen. IEEE International Conference on Pervasive Computing and Communications Workshops Proceedings. 2018
[3] K Yordanova. From Textual Instructions to Sensor-based Recognition of User Behaviour. Companion Publication of the 21st International Conference on Intelligent User Interfaces. 2016
Author Response
We sincerely appreciate your hard work and thoughtful suggestions that have greatly improved this article. Below is a point-by-point response to your comments:
1.In the literature review, when talking about graph neural networks, the authors are mixing networks applied to different input data but they do not make clear distinction between approaches applied to images and approaches applied to some other data (e.g. sensor data). As the whole literature review centers around image based AR, this sudden shift to other types of data is confusing.
Thanks for your valuable suggestions, indeed when reviewing GCN, mixing networks applied to different input data and do not make a clear distinction between approaches applied to images and approaches applied to some other data will confuse readers. We have carefully rewritten the literature review and reviewed the GCN-based method for images data that is related to the content of this article.
2.In the literature review, to make the review complete the authors should also address other types of data used for AR in industrial settings. For example there are works using sensors (e.g. see [1]) and as sensors are considered less intrusive than camera, the authors should at least mention this type of research.
Thanks for your valuable suggestions, we have carefully rewritten the literature review. Although input data from other sensors is not the focus of this article, it is undeniable that they are indeed methods for action recognition. We mentioned several representative ways in the last paragraph of section 2.1 and cited the literature you mentioned.
3.Also in the literature review, the authors talk about classical ML approaches for AR but they don't mention approaches such as HMMs, which model temporal and causal relations between the different poses and activities (e.g. see [2]). I think that such approaches are extremely relevant as there are certain causal and temporal relations between human activities especially in goal oriented scenarios (and tasks in industrial settings are usually goal oriented). I believe the authors should mention the existence of such approaches and how they differ from the way their approach utilises temporal and spatial information.
Thanks for your valuable suggestions, we have carefully rewritten the literature review. HMM, and other methods do play an important role in the research of action recognition. We added a review of HMM, AdaBoost and some other methods in the second paragraph of section 2.1.
4.Also along these lines, I wonder if the authors are aware of approaches that use high level textual instructions describing the way tasks should be executed in order to enhance the recognition performance. For example, [3] generates models of human behaviour from textual instructions such as manuals. These models are then combined with sensor data to perform activity recognition.
Thank you for your valuable suggestions. We did not have aware of approaches that use high-level textual instructions describing the way tasks should be executed in order to enhance the recognition performance. We mentioned related research in the last paragraph of 2.1 and cited the literature you mentioned.
5.In the evaluation part, the authors compare their approach with different state of the art approaches. To me the results seem very similar. The authors should perform statistical tests to test if their approach significantly outperforms the existing approaches.
Thank you for your valuable suggestions, we have detailed the previous results, as shown in Table 4, our method is better than the previous method in the accuracy of almost every category recognition. At the same time, at the beginning of December, we conducted field tests in the workshop of China UnitedPower Company, and added relevant statistical tests to the manuscript, as shown in Table 5. By statistics of the false recognition rate in actual production, we fully explained our method has practical significance and its effect is better than the existing methods.
6.- General comments:
The abstract and introduction have some language problems and need to be proofread. The language in the rest of the paper is fine.
The section numbering starts with 0 instead of with 1
We carefully revised the language problems in abstract and introduction. And we also modified the section number according to the requirements of the journal.
[1]F Moya Rueda et al. Convolutional neural networks for human activity recognition using body-worn sensors. Informatics 5 (2), 26. 2018
[2]S Whitehouse et al. Evaluation of cupboard door sensors for improving activity recognition in the kitchen. IEEE International Conference on Pervasive Computing and Communications Workshops Proceedings. 2018
[3] K Yordanova. From Textual Instructions to Sensor-based Recognition of User Behaviour. Companion Publication of the 21st International Conference on Intelligent User Interfaces. 2016

Round 2
Reviewer 1 Report
The authors have addressed the reviewers concerns. I still feel that the readability of the paper may be further improved both in terms of language and structure.
Author Response
Thank you for your suggestion, we use a professional English editing service to improve the readability of the paper and have our manuscript checked by a native English speaking. Attached is a certificate of English editing.

This manuscript is a resubmission of an earlier submission. The following is a list of the peer review reports and author responses from that submission.
Round 1
Reviewer 1 Report
The paper presents a hierarchical framework of action recognition of workers in an industrial environment using deep learning. The input video is captured by a static RGB camera. It starts with human pose estimation and then produces human skeleton motion, finally predicts action using graph convolution network method over multiple frames. The method is an end to end deep network. The paper gives a good literature review and overall the paper is well-written. Attention mechanism proved effective for action recognition. Can the method be more generalized? what if the entire human body is all equally important? instead of only the upper body. The motivation of STN is not very clear and needs clarification, particularly STN seems to improve the performance a lot in the experiment. why is it able to eliminate pose diversity? how do we know the skeleton image is adjusted to the appropriate size and pose? it'd be good to include some intermediate results to demonstrate the effectiveness of STN. The Presentation of GCN in S2.4 needs improvement and is not easy to follow. In the experiment section, the paper creates an action dataset with six operations to test the algorithm. It'd be great to include some sample images of six operations and it helps to understand the challenge of action recognition of 6 operations. It seems both [40] and the proposed method produces about the same accuracy on the dataset. Can the proposed method handle multiple people in the video? a few minor questions: Line 170 what is NAD matrix? Line 274 what is 1-distance ? Line 280 between keypoints between adjacent frames?
Reviewer 2 Report
This paper proposed a deep learning based action recognition method to automatically recognize manual operations in industrial workflows. A framework utilizing human detection and pose estimation was proposed to obtain the skeleton image. STN was used to correct skeleton images to avoid the impact of complex real-world production environments. GCD was used to extract both spatial and temporal information in skeleton images. Since human action recognition has been intensively studied and successfully applied in many scenarios, the contribution of the paper is very limited.
The word “understand” was used exchangeable with the word “recognize”. This is not how both words are used in this field. Industrial workflows/environments or not, the technology of deep learning based action recognition could be the same or similar. This reviewer suggests the authors to focus more on the technology. The author claimed a novel deep learning based method of processing video data. Using multiple deep learning networks to process video data is really not novel. It has been widely used by researchers in this field. For action recognition, studies have been conducted to take advantages of both the spatial information and temporal information of skeleton images. It is hard to identify the contribution from the authors. One can read from line 73 that “satisfactory results cannot be achieved …”. What are satisfactory results? How do you define “satisfactory”? Human action recognition has been studied for many years and was successful in many applications. Nowadays, many commercial cameras have been integrated with deep learning based action recognition algorithms to recognize human actions on factory floors. Line 425: please prove or support “our method on real production video outperform the existing research” with results. One can read from line 426 that “our study is a kind of next generation industrial and manufacturing technologies…”. This is too much for the authors to claim.